# A Mobile Sensing Framework for Bridge Modal Identification through an Inverse Problem Solution Procedure and Moving-Window Time Series Models

**DOI:** 10.3390/s23115154

**Published:** 2023-05-28

**Authors:** Mohammad Talebi-Kalaleh, Qipei Mei

**Affiliations:** Department of Civil and Environmental Engineering, University of Alberta, Edmonton, AB T6G 2H5, Canada; talebika@ualberta.ca

**Keywords:** indirect bridge health monitoring, bridge mode shape identification, moving vehicles, moving-window ARX model, inverse problem, vibration-based monitoring

## Abstract

With the rise and development of smart infrastructures, there has been a great demand for installing automatic monitoring systems on bridges, which are key members of transportation networks. In this regard, utilizing the data collected by the sensors mounted on the vehicles passing over the bridge can reduce the costs of the monitoring systems, compared with the traditional systems where fixed sensors are mounted on the bridge. This paper presents an innovative framework for determining the response and for identifying modal characteristics of the bridge, utilizing only the accelerometer sensors on the moving vehicle passing over it. In the proposed approach, the acceleration and displacement response of some virtual fixed nodes on the bridge is first determined using the acceleration response of the vehicle axles as the input. An inverse problem solution approach based on a linear and a novel cubic spline shape function provides the preliminary estimations of the bridge’s displacement and acceleration responses, respectively. Since the inverse solution approach is only capable of determining the response signal of the nodes with high accuracy in the vicinity of the vehicle axles, a new moving-window signal prediction method based on auto-regressive with exogenous time series models (ARX) is proposed to complete the responses in the regions with large errors (invalid regions). The mode shapes and natural frequencies of the bridge are identified using a novel approach that integrates the results of singular value decomposition (SVD) on the predicted displacement responses and frequency domain decomposition (FDD) on the predicted acceleration responses. To evaluate the proposed framework, various numerical but realistic models for a single-span bridge under the effect of a moving mass are considered; the effects of different levels of ambient noise, the number of axles of the passing vehicle, and the effect of its speed on the accuracy of the method are investigated. The results show that the proposed method can identify the characteristics of the three main modes of the bridge with high accuracy.

## 1. Introduction

The development of smart infrastructure has highlighted the need for automatic and real-time monitoring systems for critical transportation components such as bridges. The high cost associated with maintenance and reconstruction of these essential lifelines has made the implementation of such monitoring systems a pressing issue. Consequently, significant research has been conducted over the last few decades in the area of vibration-based monitoring of bridge structures using sensors installed directly on the bridges [1,2,3].

Previous studies have made notable contributions to bridge health monitoring. For instance, Gonzalez and Karoumi [4] proposed a damage detection method for bridges utilizing bridge weigh-in-motion (BWIM) data and machine learning techniques. Their approach employed an artificial neural network (ANN) to predict bridge health based on BWIM data, ensuring reliable evaluations over time. Similarly, Azim and Gül [5] developed a data-driven damage detection framework for truss railway bridges using operational acceleration and strain response data. Feng and Feng [6] presented a time-domain finite element (FE) model updating approach using in situ measurements of dynamic displacement responses under trainloads. Their study validated the importance of the bridge’s equivalent stiffness for accurate model updating, although extracting modal information from dynamic responses proved challenging for short-span railway bridges with high natural frequencies.

Due to the high cost of real-time monitoring of bridges using fixed sensors, recent research in structural health monitoring has explored indirect methods, called indirect bridge health monitoring (iBHM), that utilize only the vertical vibration data collected by accelerometers mounted on passing vehicles. This method, first proposed by Yang et al. [7], has recently gained significant recognition. The main objective in this research direction is to find a reliable method to determine the vertical vibration responses of desired locations on a bridge without the need for expensive fixed sensors or with minimal sensor requirements [8,9]. For example, Malekjafarian and O’Brien [10] developed a method for identifying bridge mode shapes using short time frequency domain decomposition (STFDD) of responses measured in a passing vehicle. Their approach involved segmenting the bridge and employing a multi-stage procedure using frequency domain decomposition (FDD) to estimate the mode shapes. Numerical case studies validated the performance of the method, demonstrating the accurate estimation of mode shapes under low noise levels and the presence of other traffic or signal subtraction in identical axles. Furthermore, Eshkevari et al. [11] developed novel methods for modal identification of bridges using data collected by a large number of moving sensors (vehicles). Their study proposed matrix completion methods, specifically the alternating least squares algorithm, to extract modal properties from sparse and dynamic bridge response data. Three methods were evaluated: principal component analysis, structured optimization analysis, and the natural excitation technique (NExT). The results demonstrated accurate estimations of modal properties. However, the methods had limitations in terms of computational costs, modal leakage, sparse data, user-defined points, and accuracy for higher modes. The study showcased the potential of using mobile sensor networks for bridge health monitoring and system identification. Kong et al. [12] proposed a method to efficiently extract bridge modal properties using a test vehicle composed of a tractor and trailers. They verified the method on an existing bridge, considering the effects of trailer mass and stiffness. Their findings demonstrated high visibility in extracted bridge frequencies, particularly when traffic flows provided additional excitation. However, limitations were identified, such as difficulties in accurately extracting mode shapes dominated by lateral bending due to the limited modeling of trailers.

To reduce the cost of monitoring using mobile sensing, researchers have also explored the use of smartphone sensors instead of expensive and commercially graded accelerometer sensors. Smartphone data have been widely used in different fields, such as indoor positioning [13], crime prevention [14], and even agriculture [15]. In the structural health monitoring field, smartphone sensors have also demonstrated effectiveness in various applications, such as bridge seismic monitoring [16], assessment of building damage from seismic events [17], damage detection of a 3D steel frame [18], and walking vibration analysis [19]. Experimental investigations have demonstrated the reliability of smartphone technology for bridge monitoring, particularly in identifying natural frequencies, although this area of research is still in its early stages [20]. For instance, Di Matteo et al. [21] conducted a field experiment on the Corleone bridge in Palermo, Italy, to assess smartphone-based bridge monitoring through vehicle–bridge interaction. The study successfully identified the bridge’s natural frequency using smartphone data with high accuracy. Shirzad-Ghaleroudkhani and Gül [22] developed a novel methodology for natural frequency identification of bridges using acceleration signals recorded by smartphones on passing vehicles. Their inverse filtering approach effectively removed the frequency content of the vehicle. Additionally, Sitton et al. [23] proposed postprocessing strategies to estimate a bridge’s fundamental frequency from acceleration data recorded from a traversing vehicle without prior knowledge of bridge parameters, successfully validating their approach through finite element simulations and experimental validation on a scale-model bridge.

Despite the potential benefits of mobile sensing, challenges remain in achieving accurate bridge response prediction and mode shape identification [9,24]. Complicated mathematical techniques and principles of structural dynamics are often required. Therefore, this paper aims to explore the use of mobile sensors on crossing vehicles for bridge health monitoring, leveraging their ubiquity and potential cost savings, while addressing the need for accurate modal identification and practical solutions.

To predict the bridge response using the recorded acceleration responses from a crossing vehicle, the vehicle response needs to be spatially mapped onto some virtual fixed nodes on the bridge [25]. These estimated responses can then be used to identify the dynamic characteristics and potential damages in the structure. However, due to the interpolation of the adjacent crossing axles (moving sensors), theoretical inverse problem solutions can only determine a limited part of the response signal for each fixed node on the bridge. This interpolation results in a sparse response matrix, where each row corresponds to the response of a particular fixed node and each column corresponds to the response vector of the fixed nodes in a time stamp. This matrix contains numerous missing values (invalid regions) that require advanced statistical, mathematical, or machine learning techniques to predict or complete the response signals for the virtual fixed nodes [25,26].

A few previous studies have utilized vehicle response data to identify bridge mode shapes using the sparse response data from bridges. While some of these methods have attempted to address the issue of missing values in the bridge response matrix through soft-imputing techniques [26], short time frequency domain decomposition [10], alternating least squares technique, and principal component analysis [11], these approaches often rely on engineering judgment, involve time-consuming constrained optimization processes, and require manual parameter settings. Although these limited research works have made significant contributions to drive-by modal identification, there is a need for an innovative and automated framework to overcome these limitations.

This research work presents a novel technique for identifying the modal characteristics of bridges using only accelerometer sensors mounted on vehicles passing over them. The proposed framework consists of two stages. In the first stage, an inverse problem solution approach is employed to determine the acceleration and displacement response of virtual fixed nodes on the bridge. This is achieved by using the acceleration response of the vehicle axles as input, utilizing conventional linear interpolation functions to predict the bridge displacement responses, and introducing a novel cubic spline shape function to predict the acceleration response of the assumed fixed nodes on the bridge. However, the inverse problem solution approach yields response signals with missing parts, necessitating prediction. To address this issue, a new automated moving-window time series model based on auto-regressive exogenous (ARX) techniques is proposed in this paper. In the second stage, a novel approach combines the results of singular value decomposition (SVD) on the predicted displacement responses and frequency domain decomposition (FDD) on the predicted acceleration responses. This approach allows for the accurate identification of the first mode shape and higher mode shapes of the bridge, as well as the determination of natural frequencies. The main novelties of this research lie in the utilization of a cubic spline shape function within the inverse problem solution stage to predict the acceleration response of the fixed nodes and the application of moving-window time series models to complete the predicted incomplete signals obtained from the inverse problem solution procedure. Moreover, the method distinguishes itself by identifying mode shapes of the bridge using both acceleration and displacement responses, thereby enhancing the accuracy of identification for both lower and higher modes. Numerical simulations are conducted to evaluate the effectiveness of the proposed framework, considering different levels of ambient noise, number of axles, and vehicle speed. The future implementation of the framework in smartphone sensing-based applications is also discussed.

This paper is structured as follows: Section 1 provides the introduction and review of the literature. Section 2 focuses on the background and the inverse problem solution procedure, specifically for estimating the preliminary response signals of the bridge. Section 3 introduces the details of the proposed framework, which encompasses a two-stage approach for predicting the missing parts of the response signals of the virtual fixed nodes on the bridge, as well as identifying the mode shapes and natural frequencies of the bridge. In Section 4, numerical analyses are conducted to evaluate the performance of the proposed method. The results of the analyses are presented and discussed in Section 5, where the characteristics and limitations of the proposed method are also examined. Finally, Section 6 presents the conclusions of the study, along with recommendations for future research in this area.

## 2. Background and Inverse Problem Solution Procedure

Identifying moving loads is a prevalent inverse problem in the field of structural dynamics; researchers have developed several approaches to tackle this challenge [27]. These approaches can be categorized into two main types: those relying on analytical models and those formulated using finite element models, with a specific focus on solution techniques. Another type of inverse problem encountered in vehicle–bridge dynamics pertains to the identification of structural parameters using the moving load as an excitation.

In this paper, a different approach is presented, inspired by the work of Oshima et al. [25], to address this inverse problem, utilizing a FE approach with two different shape functions to estimate the bridge response by incorporating the vehicle response as the input. The proposed approach is thoroughly discussed and successfully solved within this section, with a novel shape function being employed to enhance the accuracy of the bridge’s acceleration response.

### 2.1. Assumptions and Notations

This study assumes that the sensors are mounted on the front and rear axles of the vehicle to mitigate the effects of the suspension system [26]. For simplicity, the deformation of wheels and tires is ignored. In practice, the effect of the vehicle–bridge interaction can be eliminated by considering its empirical transfer function. Although it is assumed that the recorded data are solely accelerations, the displacements integrated twice from the accelerations will also be used as inputs for the proposed time series models. To have a linear time invariant system, the mass of the vehicle is ignored in comparison with the mass of the bridge. The speed of all traversing axles is also assumed to be identical and constant during the measurement for simplicity. The geometry of the bridge and parameters of the moving vehicle are shown in Figure 1; all other notations and variables used in this paper are introduced in Table 1.

### 2.2. Inverse Problem Solution for Bridge Response Determination Utilizing Vehicle Response

In order to determine some parts of the bridge response signals at the virtual fixed nodes utilizing the vehicle response, an inverse problem solution is first employed in this section.

Based on the principles of finite element methods, the continuous vertical displacement response of a bridge can be determined by considering a proper interpolating (shape) function and using the discrete responses of the bridge at the location of the fixed nodes (shown in Figure 1b) [28].
(1)$$y(x,t)=\left[\begin{array}{ccc} N_{1}(x) & \cdots & N_{n}(x) \end{array}\right]\left\{\begin{array}{c} y_{s_{1}}(t)\\ \vdots \\ y_{s_{n}}(t) \end{array}\right\}=N(x)D(t)$$
where **N**(*x*) is the interpolating shape function matrix and **D**(*t*) is a vector containing vertical displacements of the fixed nodes.

In Equation (1), a linear interpolating shape function is usually considered:(2)$$\begin{array}{c} \begin{array}{llllll} \;\;\;\;\;\;\;\;\;\;\;\;\;\;\;\;\;\;\;\;s_{1} & \cdots & s_{j} & \;\;\;\;\;s_{j+1} & \;\;\;\cdots & s_{n}\;\;\;\;\;\;\;\;\;\;\;\;\;\;\;\;\;\;\;\;\;\;\;\;\;\;\;\;\;\;\;\;\; \end{array}\\ N(x)=\left[\begin{array}{llllll} 0 & \cdots & \frac{x-s_{j+1}}{s_{j}-s_{j+1}} & \frac{x-s_{j}}{s_{j+1}-s_{j}} & \cdots & 0 \end{array}\right];\;\;\;\;s_{j}<x\leq s_{j+1} \end{array}$$
where *S_j_* is the location of the *j*th fixed node from the left support. *j* can be valued from 1 to (*n* − 1).

By substituting the location history of the moving axles (*x*_1_(*t*) to *x*_m_(*t*)) in Equation (1), the nodal displacement responses of the moving axles can easily be extracted [26].
(3)$$Y(t)=\left\{\begin{array}{c} y_{1}^{v}(t)\\ \vdots \\ y_{m}^{v}(t) \end{array}\right\}=\left[\begin{array}{c} N(x_{1}(t))\\ \vdots \\ N(x_{m}(t)) \end{array}\right]\left\{\begin{array}{c} y_{s_{1}}(t)\\ \vdots \\ y_{s_{n}}(t) \end{array}\right\}=N_{v}(t)D(t)$$
where *y_i_^v^(t*) is the vertical displacement response of the *i*th axle of the vehicle and **N***_v_*(*t*) is an interpolating shape function matrix for estimating the response of the moving axles.

For a general condition of *m ≠ n*, multiplying Equation (3) by the pseudoinverse of the matrix **N***_v_*(*t*) produces the nodal displacements of the bridge as a function of displacements of the moving axles.
(4)$$D(t)=\left({\left(N_{v}^{tr}(t)N_{v}^{}(t)\right)}^{-1}N_{v}^{tr}(t)\right)Y(t)$$

Taking the second derivation from both sides of the previous equation will produce a similar relation between the acceleration responses.
(5)$$\ddot{D}(t)=\left({\left(N_{v}^{tr}(t)N_{v}^{}(t)\right)}^{-1}N_{v}^{tr}(t)\right)\ddot{Y}(t)$$

### 2.3. Valid Regions of the Estimated Signals

Although in Equations (4) and (5), the pseudoinverse of the **N***_v_*(*t*) matrix is multiplied in all the responses of the moving axes, based on our numerical observations, the use of the pseudoinverse of the **N***_v_*(*t*) matrix based on the assumption of either linear or spline shape function, which is introduced in the next part, leads to an accurate estimation only for the responses of the nodes located in the vicinity of the moving axles at that time stamp; in time intervals outside of that, the prediction error will be very large, which cannot be used for structural health monitoring applications. The main reason for the error is the fact that the matrix **N***_v_*(*t*) is a sparse matrix and has lots of zeros outside of the valid region of the response, thereby its inverse can result in large errors. Although the proposed rule to determine the valid regions of the estimated response is more general and can be applied for any number of axles, Figure 2 illustrates the procedure for a case of the three moving axles crossing the bridge at an arbitrary time stamp such as *t*. Since there are at least two moving axles in the vicinity of the nodes *S_j_*_−1_ and *S_j_* to contribute to their response in the given time, the estimated response by the theoretical method for these two nodes at time *t* is considered valid. The main reason for adopting this assumption is that, in order to determine the response of a fixed node at time *t* using the inverse solution of Equation (3), there must be at least two non-zero rows corresponding to that fixed node in the matrix **N***_v_*(*t*). The inverse of small or zero values in invalid regions approaches infinity and results in high error issues.

Considering this approach, the expected valid and invalid regions for the responses of each fixed node can be determined. Figure 3 shows a schematic visualization of the valid and invalid data regions in the matrix of nodal responses. It should be noted that, if the number of axles increases, the valid region of matrix **D** will increase. Therefore, the missing ratio of the matrix will be reduced. Furthermore, the more fixed nodes that are considered, the shorter valid regions and the higher missing rate we will have in matrix **D**.

## 3. Proposed Response Prediction and Modal Identification Methodology

The proposed framework from collecting the acceleration response of the crossing axles to identifying the modal characteristics of the bridge is summarized in Figure 4. In this study, an inverse problem solution procedure is employed to estimate the initial displacement response signals of the bridge. Initially, a linear shape function is utilized for this purpose. However, based on numerical investigations, it has been demonstrated that incorporating a cubic spline shape function in the inverse solution procedure yields more precise results for bridge acceleration responses. The subsequent sub-section presents a detailed description of the proposed technique, which involves the use of a cubic spline interpolation function within the inverse problem solution procedure.

### 3.1. Continuous Cubic Spline as the Shape Function

A cubic spline is a piecewise polynomial in which the coefficients of each polynomial are fixed between joints [29]. In this paper, an innovative approach to estimate the bridge nodal responses at the valid regions is proposed. The novelty of the proposed approach is utilizing cubic spline polynomials as the shape function for the displacement field in lieu of the conventional discontinuous linear ones (Figure 5). Although a cubic spline interpolator is a continuous combination of some piecewise nonlinear cubic polynomials, it is shown that the linearity between the nodal responses and the response function will still be valid; the relation between the nodal responses and response field function can be written in the form of Equation (1). 

It should be noted that natural spline is employed in this paper due to the fact that the first and last supports of bridges are generally roller or hinge type, which releases the moment reaction at the supports and results in zero curvature at those points (i.e., **N**″(*x* = 0) and **N**″(*x* = *L*) are considered zero).
(6)$$N(x)={\hat{D}}_{j,:}{\left(x-s_{j}\right)}^{3}+{\hat{C}}_{j,:}{\left(x-s_{j}\right)}^{2}+{\hat{B}}_{j,:}(x-s_{j})+{\hat{A}}_{j,:}\;\;\;\;;\;\;\;\;s_{j}<x\leq s_{j+1}$$
where the matrix of coefficients $\hat{A}$, $\hat{B}$, $\hat{C}$, and $\hat{D}$ are the size of *n* by *n* and to be calculated via the following equations and the row index, *j*, can be valued from 1 to (*n* − 1) [29]:(7)$$\hat{A}=I_{n}$$
(8)$$\hat{C}=GH^{-1}$$
(9)$$\hat{B}=\frac{1}{\Delta s}\left[{\hat{A}}_{2:n,:}\;-{\hat{A}}_{1:n-1,:}\;\right]-\frac{\Delta s}{3}\left[{\hat{C}}_{2:n,:}\;+2{\hat{C}}_{1:n-1,:}\;\right]$$
(10)$$\hat{D}=\frac{1}{3\Delta s}\left[{\hat{C}}_{2:n,:}\;-{\hat{C}}_{1:n-1,:}\;\right]$$
(11)$$G=\frac{3}{\Delta s}\left[\begin{array}{lllllll} 0 & 0 & 0 & 0 & \cdots & \cdots & 0\\ 1 & -2 & 1 & 0 & \cdots & \cdots & 0\\ 0 & 1 & -2 & 1 & \cdots & \cdots & 0\\ \vdots & \vdots & \vdots & \vdots & \ddots & \ddots & \vdots \\ 0 & 0 & 0 & \cdots & 1 & -2 & 1\\ 0 & 0 & 0 & \cdots & 0 & 0 & 0 \end{array}\right]$$
(12)H=Δs1Δs000⋯⋯01410⋯⋯00141⋯⋯0⋮⋮⋮⋮⋱⋱⋮000⋯141000⋯001Δs

In the previous equations, Δ*s* is the distance between two adjacent virtual fixed nodes that can be assumed constant.

After the determination of **N**(*x*), the continuous response function of the bridge can be calculated by Equation (1).

### 3.2. Proposed Moving-Window ARX Model to Complete the Missing Parts of the Estimated Responses

As discussed in Section 2, the classical method for solving the inverse problem can only estimate a small portion of the bridge displacement signal using drive-by data. In order to have the complete response for all the fixed nodes, a signal forecasting and completion approach is required.

In the present research, an innovative moving-window forecasting framework based on auto-regressive time series models with exogenous input (ARX) is introduced. The main motivation for this procedure is that the short valid part of the estimated response signal for a given fixed node can be trained by the corresponding parts of the responses of the adjacent nodes to forecast the missing response of the given node. 

In other words, the proposed approach considers a unique ARX model for each of the fixed nodes. Therefore, (*n* − 2) different ARX models can be established for the whole system. The proposed procedure can be utilized to forecast the missing parts of the signal in both backward and forward directions; however, it should be noted that, for training and predicting the missing parts in the backward direction, the reverse of the signals is used (see Figure 6).

The ARX model structure is generally given by the following equation [30]:(13)$$y(t)+a_{1}y(t-\Delta t)+\;\;\cdots \;\;+a_{na}y(t-na\Delta t)=a_{1}u(t-nk)+\;\;\cdots \;\;+a_{nb}u(t-\left(nb+nk-1\right)\Delta t)+e(t)$$
where *y*(*t*) and *u*(*t*) are the output and the input of the system, respectively; $a_{1}$, …, $a_{na}$ and $b_{1}$, …, $b_{nb}$ are parameters of the model, which can be identified through the least-squares optimization approach; *e*(*t*) is a white-noise disturbance value. 

The main reason for considering the ARX models is that the responses of two different fixed nodes located on the bridge can be decomposed to modal response components utilizing modal expansion principles for *n_d_* modes [31]:(14)$$y_{s_{j}}(t)=\left[\begin{array}{ccc} \phi_{s_{j},1} & \cdots & \phi_{s_{j},n_{d}} \end{array}\right]\left\{\begin{array}{c} q_{1}(t)\\ \vdots \\ q_{n_{d}}(t) \end{array}\right\}$$
(15)$$D\left(t\right)=\left\{\begin{array}{c} y_{s_{1}}(t)\\ \vdots \\ y_{s_{n}}(t) \end{array}\right\}=\left[\begin{array}{ccc} \phi_{s_{1}1}^{} & \cdots & \phi_{s_{1}n_{d}}^{}\\ \vdots & \ddots & \vdots \\ \phi_{s_{n}1}^{} & \cdots & \phi_{s_{n}n_{d}}^{} \end{array}\right]\left\{\begin{array}{c} q_{1}(t)\\ \vdots \\ q_{n_{d}}(t) \end{array}\right\}=\Phi^{s}Q(t)$$

In a general case, multiplying both sides of Equation (15) by the pseudoinverse of **Ф***^s^* and substituting the resulting **Q**(*t*) in Equation (14) yields to Equation (17):(16)$$Q(t)=\left\{\begin{array}{c} q_{1}(t)\\ \vdots \\ q_{n_{d}}(t) \end{array}\right\}={\left[\begin{array}{ccc} \phi_{s_{1}1}^{} & \cdots & \phi_{s_{1}n_{d}}^{}\\ \vdots & \ddots & \vdots \\ \phi_{s_{n}1}^{} & \cdots & \phi_{s_{n}n_{d}}^{} \end{array}\right]}^{-1}\left\{\begin{array}{c} y_{s_{1}}(t)\\ \vdots \\ y_{s_{n}}(t) \end{array}\right\}$$
(17)$$y_{s_{j}}(t)=\left[\begin{array}{ccc} \phi_{s_{j},1} & \cdots & \phi_{s_{j},n_{d}} \end{array}\right]{\left[\begin{array}{ccc} \phi_{s_{1}1}^{} & \cdots & \phi_{s_{1}n_{d}}^{}\\ \vdots & \ddots & \vdots \\ \phi_{s_{n}1}^{} & \cdots & \phi_{s_{n}n_{d}}^{} \end{array}\right]}^{-1}\left\{\begin{array}{c} y_{s_{1}}(t)\\ \vdots \\ y_{s_{n}}(t) \end{array}\right\}=\left[\begin{array}{ccc} b_{j,1} & \cdots & b_{j,n} \end{array}\right]\left\{\begin{array}{c} y_{s_{1}}(t)\\ \vdots \\ y_{s_{n}}(t) \end{array}\right\}$$

According to our numerical investigations, only the contribution of the adjacent nodes is high for the response prediction of the *j*th node. Therefore, we neglect the contributions of the other fixed nodes in Equation (17) and, by doing so, the pattern of the response can still be maintained. 

Comparing Equation (17) with the general ARX model introduced in Equation (13), the time series model for the *j*th node can be simplified as follows:(18)$$y_{s_{j}}(t)+\;\;a_{j,1}y_{s_{j}}(t-\Delta t)+\;a_{j,2}y_{s_{j}}(t-2\Delta t)=b_{i,1}y_{s_{j+1}}(t)+\;\;b_{j,2}y_{s_{j+1}}(t-\Delta t)+\;b_{j,3}y_{s_{j+1}}(t-2\Delta t)+e(t)$$

As obtained in Equation (18), the information on external force and vehicle characteristics in the proposed model is not required. The rationale behind considering a second-order ARX model is that using acceleration response in the time series model can give better accuracy. 

By performing the proposed method for all the intermediate nodes, some parts of the bridge response signals can be completed according to Figure 7. As can be seen from the figure, this approach is only able to complete the missing parts of the signal in which the input for the time series model can be provided, considering the missing gap between the two consecutive signals.

In order to complete the entire responses of the fixed nodes, the moving ARX algorithm will be applied through a straight-forward iterative procedure. The proposed signal completion framework is shown in Figure 8 for the displacement response of an intermediate fixed node. It is important to highlight that the number of iterations needed depends on the quantity of moving axles and the number of virtual fixed nodes. For a model subjected to three-axle moving loads, a total of six iterations are required. As the number of axles increases, the number of iterations decreases due to a reduction in the valid regions. During each iteration of the algorithm, the predicted regions are combined with the initial valid regions, leading to an expansion of the valid regions within the response matrix. The iterative process continues until there are no remaining invalid regions in the response matrix.

### 3.3. Integrating Displacement and Acceleration Responses for Modal Identification

In this paper, a novel mode shape identification through the response of the moving wheels only is considered. To be more precise, both displacement and acceleration of the fixed nodes are first estimated with the aid of the proposed moving ARX method, then singular value decomposition (SVD) is applied on all of the nodal displacement responses to identify the first mode shape; however, for the higher modes and identification of natural frequencies, frequency domain decomposition (FDD) is utilized, considering the acceleration responses of the fixed nodes. This is based on the fact that combining displacement and acceleration responses in modal identification improves accuracy and reliability. Displacement responses are effective for identifying lower modes, while acceleration responses capture high-frequency components and higher modes more accurately [32]. The mathematical relationship between accelerations and displacements reduces high-frequency components in dynamic displacements. This reduction is due to the fact that the integration operation acting as a low-pass filter and the accumulation of the area under the acceleration curve during integration [33].

Since the SVD can extract the orthogonal vectors of an arbitrary matrix, it can be applied to the completed response matrix (**D**) to determine **Ф***^s^*:(19)$$D=U\Sigma V^{tr}$$
where **U** and **V** are composed of the left and right singular vectors of matrix **D**, respectively; they are orthogonal matrices. **Σ** is a diagonal matrix containing singular values of **D**.

By comparing the right side of Equations (15) and (19), identification of the mode shape can be performed with high accuracy [25]:(20)$$\Phi^{s}=U\Sigma^{{}^{\frac{1}{2}}}$$

We observed that SVD can identify the first mode shapes with high accuracy if applied on the displacement response matrix of the bridge.

As mentioned earlier, in order to identify the mode shapes through the acceleration response matrix of the bridge, the FDD method is employed [34]. The basis of FDD is presented in the following paragraph.

From statistics, the correlation matrix between the response of the fixed nodes can be constructed using Equation (21):(21)$$R_{\ddot{y}\ddot{y}}(\tau )=\frac{1}{T}\int_{0}^{T}\ddot{D}(t){\ddot{D}}^{tr}(t-\tau )dt=\left[\begin{array}{ccc} R_{{\ddot{y}}_{s_{1}}{\ddot{y}}_{s_{1}}}\left(\tau \right) & \cdots & R_{{\ddot{y}}_{s_{1}}{\ddot{y}}_{s_{n}}}\left(\tau \right)\\ \vdots & \ddots & \vdots \\ R_{{\ddot{y}}_{s_{n}}{\ddot{y}}_{s_{1}}}\left(\tau \right) & \cdots & R_{{\ddot{y}}_{s_{n}}{\ddot{y}}_{s_{n}}}\left(\tau \right) \end{array}\right]$$

Considering modal expansion and substituting Equation (15) in Equation (21) gives:(22)$$R_{\ddot{y}\ddot{y}}\left(\tau \right)=\frac{1}{T}\int_{0}^{T}\Phi \ddot{Q}(t){\ddot{Q}}^{tr}(t-\tau )\Phi^{tr}dt=\Phi R_{\ddot{q}\ddot{q}}(\tau )\Phi^{tr}$$

Then, taking Fourier transform from both sides of the latter equation produces the matrix containing cross/auto power spectrums of the response signals:(23)$$G_{\ddot{q}\ddot{q}}(\omega_{i})=\Phi G_{\ddot{q}\ddot{q}}(\omega_{i})\Phi^{tr}$$
(24)$$G_{\ddot{q}\ddot{q}}(\omega_{i})=U_{i}\Sigma_{i}V_{i}^{tr}$$

As can be understood from Equations (23) and (24), the SVD of matrix $G_{\ddot{q}\ddot{q}}(\omega_{i})$ should be calculated for each frequency, *ω_i_*, in which the matrix of singular values (**Σ***_i_*) is a diagonal matrix containing modal FRFs and each column of the matrix of singular vectors (**U***_i_* and **V***_i_*) represents the mode shapes corresponding to the given frequency *ω_i_*.

## 4. Results

### 4.1. Model Setup

To evaluate the effectiveness of the proposed framework, a comprehensive numerical analysis is conducted on a single-span simply supported bridge using the finite element software package, ABAQUS. The bridge under consideration has a span length of 40 m and a rectangular cross-section with dimensions of 3 m wide and 1.5 m high. The material properties of the bridge are assigned based on concrete, with a density of 2400 kg/m^3^ and an elastic modulus of 27.5 GPa.

In the numerical model, the bridge is subjected to the loading of a three-axle moving vehicle. The distance between the axles is set to 2.5 m, as shown in Figure 1. The natural frequencies of the bridge model are determined to be 1.44 Hz, 5.76 Hz, and 12.95 Hz for the first three modes, respectively.

A constant speed of 60 km/h is assigned to all the moving axles. The analysis is performed using a linear implicit dynamic analysis approach, considering the contacts between the moving axles and the bridge. The simulation is terminated when the foremost moving axle reaches the right end of the bridge.

For data acquisition, the accelerometers mounted on the moving vehicle have a constant sampling frequency of 200 Hz. Although a total of nine virtual fixed nodes are defined on the bridge, for the purpose of verification and comparison, only three specific fixed nodes located at ¼, ½, and ¾ of the span are selected to verify the displacement and acceleration responses.

Further details regarding the numerical model can be found in the previous work of the authors [26]. The established numerical setup provides a realistic representation of a bridge structure and allows for the comprehensive evaluation of the proposed framework’s performance.

### 4.2. Interpretation of Results

As explained in Section 2, by utilizing the measured acceleration of the moving axles in Equation (5), the valid part of the acceleration response signal of the fixed nodes on the bridge can be estimated. Similarly, to determine the displacement response signal, it is enough to double integrate the measured acceleration signals of the axles and put them in Equation (4). 

In order to evaluate the proposed framework, the exact response of the fixed nodes on the bridge will be used directly from the numerical model. Although the acceleration and displacement responses of all nine fixed nodes can be determined using the proposed method, only the results from the linear and spline shape functions of three validation nodes are shown in Figure 9 and Figure 10.

The displacement and acceleration responses of the fixed nodes are determined using linear and spline shape functions, respectively, as outlined earlier. Figure 9 shows that the cubic spline shape function proposed in this study provides more accurate acceleration response estimates of the fixed nodes in the valid regions compared with the conventional linear approach. On the other hand, the linear shape function provides more precise displacement response estimates compared with the cubic spline shape function (Figure 10).

The difference observed between the linear and spline shape functions, regarding their impact on displacement response estimates, can be attributed to multiple factors. Firstly, the inherent characteristics of the displacement response itself play a significant role. Displacement responses primarily consist of low-frequency components that reflect the overall steady-state behavior of the system. The linear shape function, with its linear interpolation, aligns well with this low-frequency behavior, resulting in more precise displacement estimates.

Secondly, acceleration responses exhibit more complex dynamics and transient behaviors, often characterized by high-frequency oscillations. The cubic spline shape function, which incorporates higher-order interpolation, is better equipped to capture these intricate features, leading to more accurate estimates in the acceleration domain.

In summary, the choice between the linear and the spline shape functions depends on the specific nature of the response being analyzed. The linear shape function excels in capturing low-frequency displacement components, while the cubic spline shape function is advantageous for accurately representing the complex dynamics and high-frequency oscillations present in acceleration responses.

Hence, both responses of the fixed nodes are utilized in the proposed modal identification method. It is worth noting that higher accuracy of the determined responses in the valid regions results in higher accuracy of the predicted whole signals from the proposed moving time series model, based on the authors’ numerical observations.

The predicted displacement and acceleration responses of the bridge at the verification points, using only the measured acceleration of the moving axles and their relative amplitude errors, are presented in Figure 11 and Figure 12, respectively. The relative error of the predicted responses outside the valid regions is high in the case of a three-axle vehicle. However, as shown in the following sections, using more moving axles reduces these errors and, for all models, the mode shape identification accuracy is very high.

It is well known that the identification of lower modes of structures is easier using displacement responses, while higher modes can be identified using acceleration responses with higher accuracy [33]. Accordingly, a hybrid mode shape identification framework is proposed, where SVD is applied to the predicted displacement responses of the fixed nodes (based on the linear shape function) to identify the first mode shape. On the other hand, the FDD technique is employed to identify the higher modes and natural frequencies by analyzing the acceleration responses (based on the cubic spline shape function) of the fixed nodes.

The results of the identified mode shapes and natural frequencies are presented in Figure 13 and Table 2, respectively. It is worth noting that the identified natural frequencies are based on the completed acceleration responses and considering the plot of the first singular value obtained from FDD.

## 5. Discussion

### 5.1. Sensitivity of the Inverse Solution to the Number of Virtual Fixed Nodes

The accuracy of the theoretical inverse solution method in Equations (4) and (5) is highly dependent on the number of virtual fixed nodes, which is equivalent to the mesh size of the bridge element. Therefore, it is crucial to investigate the sensitivity of the estimated response in the valid regions for various numbers of fixed nodes using both linear and spline shape functions. As shown in Figure 14, for the linear shape function case, increasing the number of fixed nodes improves the accuracy of the estimated acceleration and displacement responses of the mid-span point up to a certain point. Beyond this point, the accuracy of the estimated response declines steadily. The ascending branch of the curve can be attributed to the fact that the finite element method requires an increased number of intermediate nodes (number of elements) to determine the displacements of the fixed nodes with higher accuracy. However, increasing the number of fixed nodes leads to an increase in the number of columns in matrix **N**(*t*) and, consequently, a large computational error in calculating the pseudoinverse of **N**(*t*). The second part of the sensitivity graph is downward and indicates a decrease in the accuracy of the estimated response. In contrast to the linear shape function results, the cubic spline shape function can achieve high accuracy with even a few interpolating fixed nodes, such as three points, but it is more sensitive to an increase in the number of fixed nodes. To ensure a fair comparison between the two methods, this study employs a constant value of nine fixed nodes on the bridge. It is worth noting that the coefficient of determination, R^2^, is used in the subsequent figures to evaluate the similarity between two signals as well as the fitting accuracy.

### 5.2. Influence of the Number of Moving Axles

As previously discussed, a larger number of axles passing over the bridge results in a longer valid region for the response signals. This provides a better measurement of the accuracy of the preliminary model fit and is expected to enhance the accuracy of the bridge response prediction. To evaluate this hypothesis, three different types of moving vehicles, each with four, six, or 8 axles, were considered; the proposed method was used to determine the displacement, acceleration response of the fixed nodes, and modal characteristics of the bridge.

Figure 15 and Figure 16 show the predicted displacement and acceleration responses, respectively, for the mid-span of the bridge, along with their time distribution of the prediction error relative to the exact response. As anticipated, the relative prediction error of the mid-span response was significantly reduced with an increase in the number of axles. Figure 17 shows the three main identified mode shapes based on the predicted acceleration and displacement responses of all fixed nodes for all three different loading types. It can be observed that, although the accuracy of identifying higher mode shapes increased with an increase in the number of axles, the first mode shape was not significantly affected.

It is important to note that an increase in the number of axles would require a corresponding increase in the number of sensors, resulting in additional costs for monitoring the structure. However, there was no significant change in the accuracy of the identified modal characteristics. Therefore, using moving vehicles with fewer axles could reduce the costs of bridge health monitoring while maintaining an acceptable level of accuracy.

### 5.3. Influence of Vehicle Speed on the Identification Results

The speed of the vehicles passing over the bridge is one of the main parameters that may affect the accuracy of identification based on the vehicle response. In this section, the effect of the parameter on the proposed method is evaluated. The speed of the moving axles is varied between 20 and 80 km/h and its effect on the accuracy of the identified mode shapes and natural frequencies is investigated. Figure 18 compares the modal assurance criterion (MAC) values of the first three identified mode shapes at different speeds for three different vehicles with varying numbers of axles. Similarly, Figure 19 shows the relative error in identifying the natural frequencies for the first three modes at different speeds and for different numbers of axles.

Upon analyzing these figures, it can be inferred that the accuracy of mode shape identification is generally higher at lower speeds, since more time information can be obtained from the bridge response. However, the accuracy of identifying the natural frequencies reduces by almost half as the vehicle speed increases. Furthermore, increasing the number of moving axles does not significantly affect the accuracy of the identified modal characteristics through the proposed hybrid method, where the predicted displacement responses are considered for the first mode and the acceleration responses are used for the higher modes.

### 5.4. Investigation of Ambient Noise Effects

Measurement errors and environmental vibrations can affect the accuracy of the proposed framework. To investigate the effects of ambient noise on the identification of mode shapes and natural frequencies using the proposed technique, different levels of artificial noise were added to the measured acceleration response of the moving axles assuming a zero-mean Gaussian distribution. The applied noise amplitude was considered as a percentage of the RMS of the measured acceleration in the range of 1–5% (corresponding to the signal-to-noise ratio of 40–26 dB).

Figure 20 shows the MAC values of the identified mode shapes by the proposed hybrid method in different levels of ambient noise compared with their exact values. The presence of ambient noise reduces the accuracy of the mode shape identification, although this sensitivity to noise is more evident in some cases, such as for six moving axles. Moreover, although the sensitivity of the first mode detection in different levels of noise has decreased with an increase in the number of axles, this trend is almost inverted for higher modes.

It should be noted that the difference in the behavior pattern between the first mode and the higher modes is due to the use of different methods. As mentioned in Section 3, the SVD method is used to identify the shape of the first mode from the predicted displacement responses of the bridge, while the FDD method is used to identify the higher modes from the acceleration responses in this study; their behavior is also different in different noise levels. In conclusion, the hybrid identification technique can detect the mode shapes reasonably accurately for all three modes while utilizing fewer axles.

Figure 21 depicts the relative error of the first three identified natural frequencies for the bridge compared with their exact values at different levels of ambient noise. The results indicate that the proposed method has high accuracy in determining the natural frequencies of the higher modes and is robust to noise. However, this behavior is not observed when investigating the effect of noise on the identified frequency of the first mode through the predicted acceleration response signal for the fixed nodes. The main reason for this is that the FDD method cannot accurately determine the position of the first peak of the first singular value diagram (first mode), utilizing the acceleration results due to the presence of ambient noise. The accuracy and robustness of the first identified natural frequency is generally higher in models with a smaller number of moving axles, highlighting the high capability of the proposed method for implementing it using the response of conventional vehicles passing over the bridge.

Overall, the average detection error at different levels of ambient noise for the frequency of the first mode shape is less than 10%, while for the frequency of higher modes it is less than 5%. These results indicate the promising efficiency of the proposed framework, which utilizes only the response of the axles of the moving vehicle, in identifying the mode shapes and natural frequencies of the bridge, even in the presence of ambient noise.

## 6. Conclusions

This paper contributes to indirect bridge health monitoring through the introduction of an automated and comprehensive framework based on an inverse problem solution approach and a novel moving-window time series model for response prediction. The major findings and contributions of this research can be summarized as follows:Accurate modal characteristics’ identification: The proposed method demonstrates accurate identification of the modal characteristics for the first three modes of bridges within normal traffic speeds ranging from 20 to 70 km/h. This capability is crucial for assessing the structural health and integrity of bridges.Novel use of cubic spline shape function and moving-window time series models: The research introduces the use of a cubic spline shape function within the inverse problem solution for predicting acceleration responses. Additionally, the application of moving-window time series models to complete the predicted signals further enhances the accuracy of the framework.Novel approach for mode shape identification: The framework utilizes predicted displacement and acceleration responses to identify the first and higher mode shapes of the bridge, respectively. This approach enhances the accuracy and robustness of mode shape identification for lower and higher modes.Cost-effective solution: The method requires only one vehicle with a limited number of axles, which significantly reduces the number of sensors compared with traditional fixed sensor setups. This offers a cost-effective solution for bridge health monitoring without compromising accuracy.

However, there are certain limitations that should be acknowledged:Influence of the number of moving axles: The accuracy of identification improves with an increase in the number of axles passing over the bridge. However, there is no significant effect on the identification of the first mode shape. This finding highlights the need to optimize the number of axles (vehicles) used in bridge monitoring systems.Influence of vehicle speed on identification: Mode shape identification is more accurate at lower speeds, while the accuracy of identifying natural frequencies decreases with higher vehicle speeds. Considering vehicle speed is important for designing effective bridge monitoring strategies.Sensitivity to ambient noise: As expected, the presence of ambient noise reduces the accuracy of mode shape identification, particularly for the higher modes. The accuracy and robustness of the first identified natural frequency is generally higher in models with fewer moving axles.

Further research and improvements can be pursued in the following areas:Enhancement of the response prediction models: exploring different models or techniques to improve the accuracy of predicted responses can lead to more precise identification of modal characteristics and better performance in the presence of noise.Multi-vehicle scenarios: investigating the applicability of the proposed method in scenarios with multiple vehicles of varying speeds crossing the bridge can provide a more comprehensive understanding of its capabilities and limitations.Experimental validation: conducting experimental investigations on real-life structures will be crucial to validate the accuracy and effectiveness of the proposed framework in practical applications.

In summary, while the proposed framework presents a promising approach for indirect bridge health monitoring, further research is needed to address the limitations and to refine the method for broader applicability and improved accuracy in real-world scenarios. These advancements will contribute to the field of bridge health monitoring, ensuring the safety and longevity of transportation infrastructure.

## Figures and Tables

**Figure 1 sensors-23-05154-f001:**
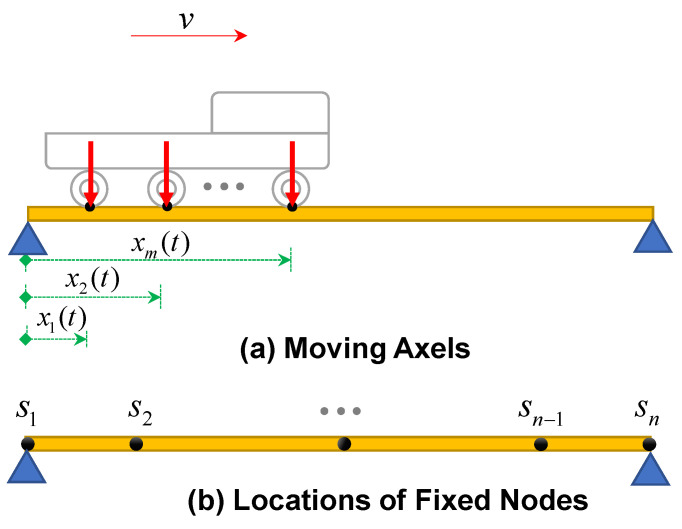
Illustration of the moving axles and fixed nodes.

**Figure 2 sensors-23-05154-f002:**
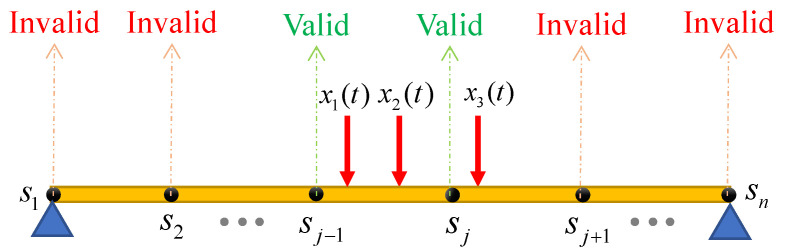
Definition of the valid and invalid regions of the estimated nodal displacements.

**Figure 3 sensors-23-05154-f003:**
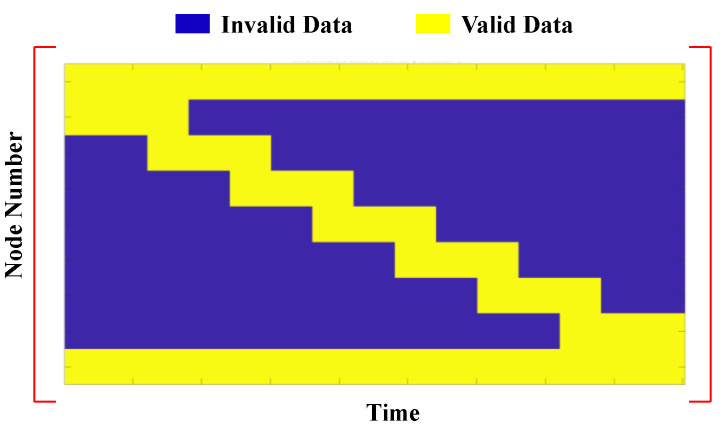
Visualization of valid and invalid regions in matrix **D**.

**Figure 4 sensors-23-05154-f004:**
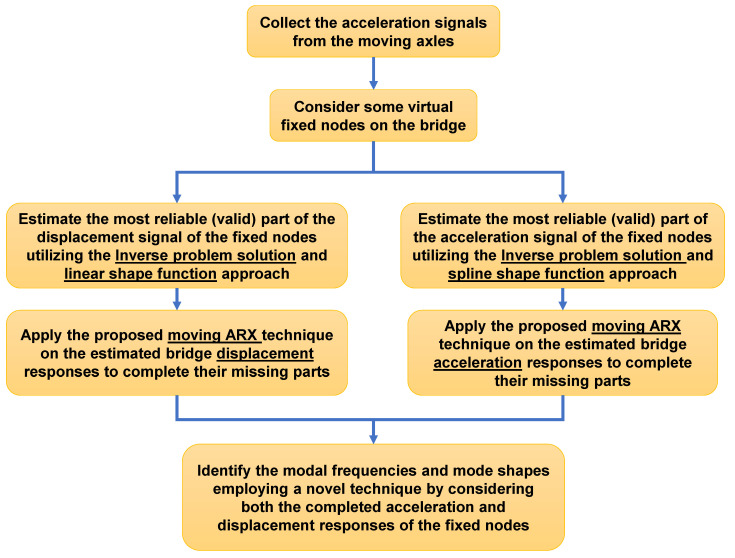
Flow chart of the proposed drive-by modal identification technique.

**Figure 5 sensors-23-05154-f005:**
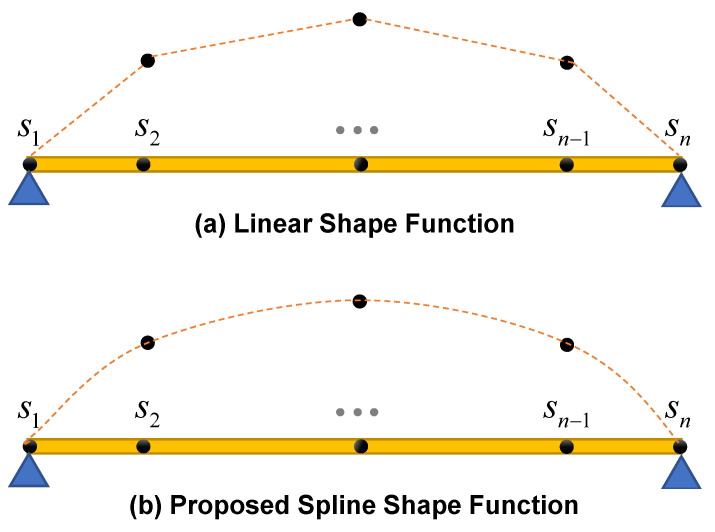
Linear vs. the proposed cubic spline shape function.

**Figure 6 sensors-23-05154-f006:**
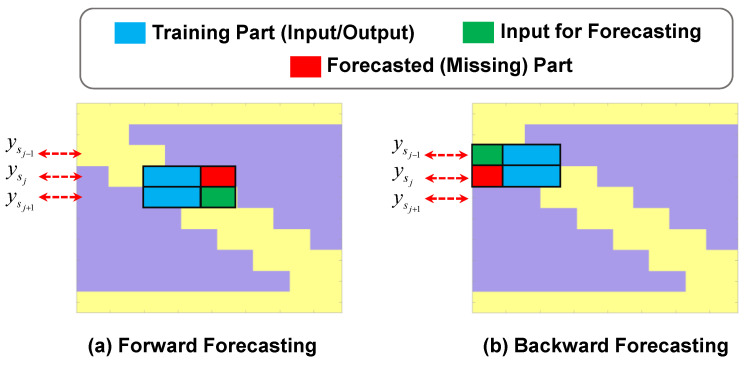
Explanation of the training parts and input in the ARX model for forecasting the missing parts of the response signal of node *s_j_*.

**Figure 7 sensors-23-05154-f007:**
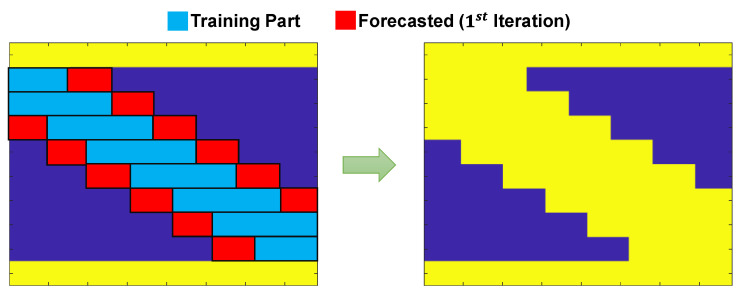
The first iteration of the proposed moving ARX technique to complete the missing parts of the response matrix, where the inputs are available for forecasting backward and forward values using the time series models.

**Figure 8 sensors-23-05154-f008:**
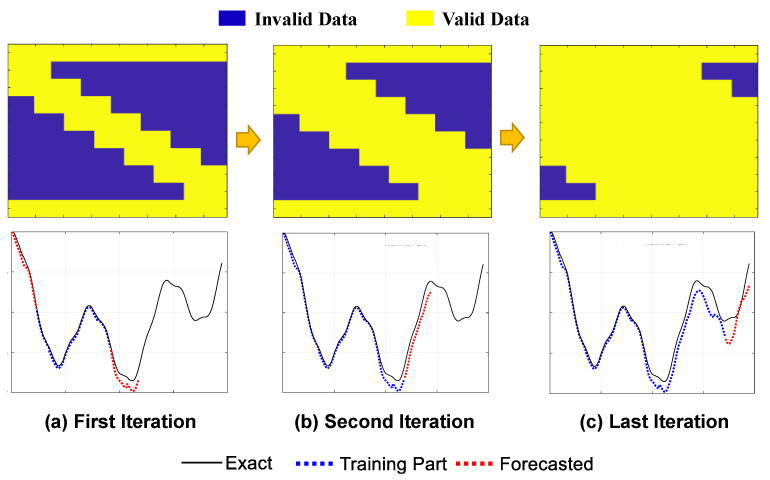
An illustrative example of the iterative moving ARX framework to complete the response signal of fixed nodes (applicable for either displacement or acceleration).

**Figure 9 sensors-23-05154-f009:**
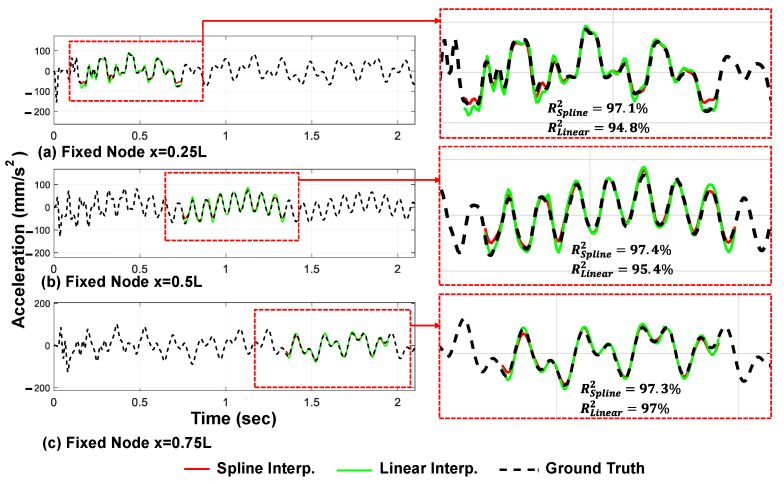
Estimated acceleration responses of the bridge in their valid regions through the linear and spline shape functions and inverse problem solution (three moving axles).

**Figure 10 sensors-23-05154-f010:**
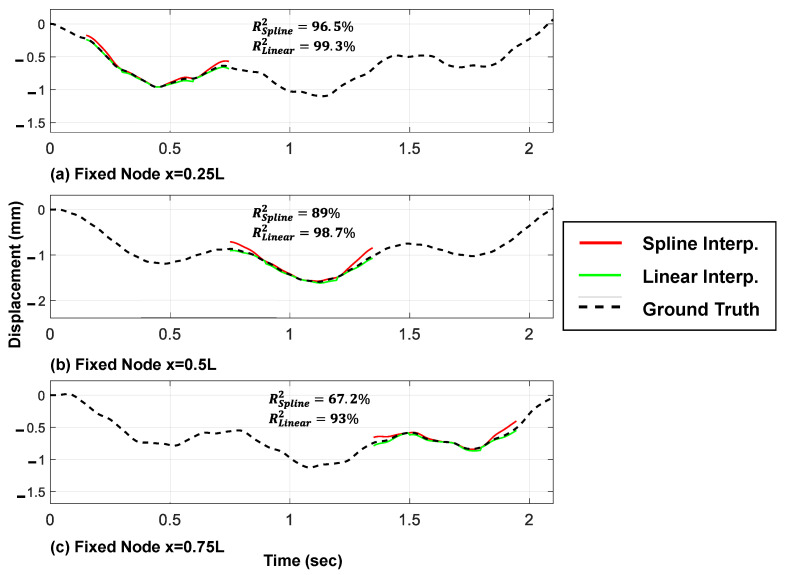
Estimated displacement responses of the bridge in their valid regions through the linear and spline shape functions (three moving axles).

**Figure 11 sensors-23-05154-f011:**
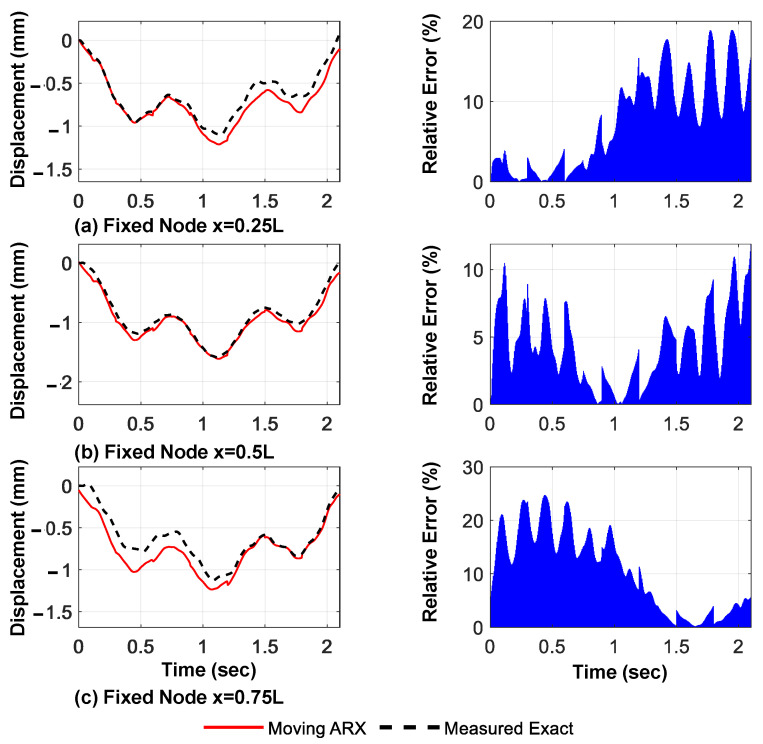
Predicted displacement responses extracted using the proposed framework and their relative amplitude errors (three moving axles, linear shape function).

**Figure 12 sensors-23-05154-f012:**
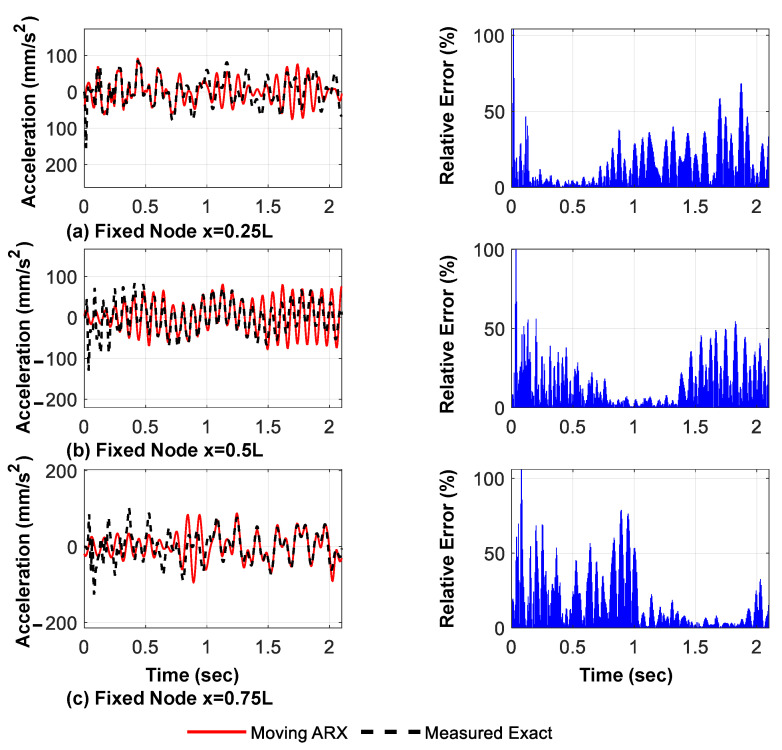
Predicted acceleration responses extracted using the proposed framework and their relative amplitude errors (three moving axles, cubic spline shape function).

**Figure 13 sensors-23-05154-f013:**
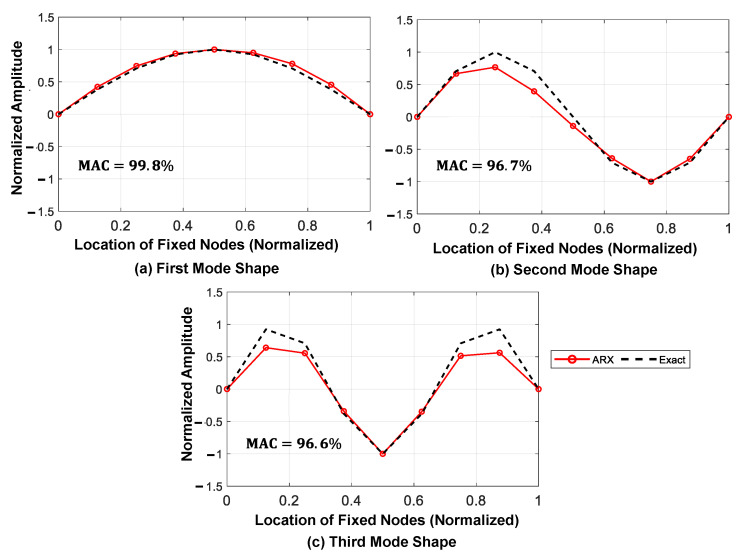
Identified mode shapes of the bridge using the proposed framework (three moving axles).

**Figure 14 sensors-23-05154-f014:**
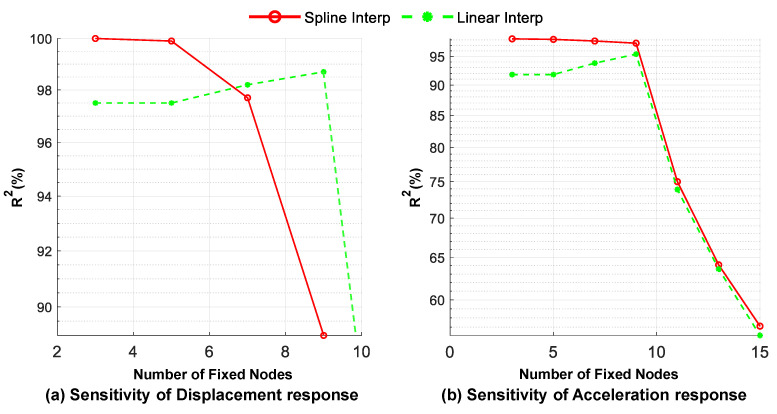
Sensitivity of the estimated responses using the inverse solution for the mid-span point in the valid region to the number of fixed nodes (three moving axles).

**Figure 15 sensors-23-05154-f015:**
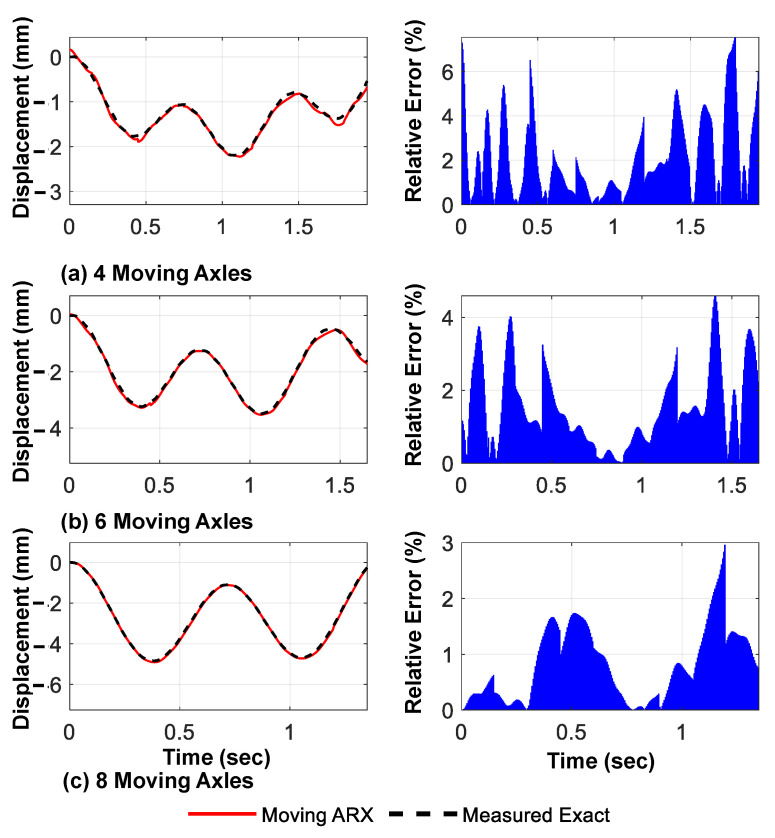
Predicted displacement response of mid-span using the proposed framework and their relative to the amplitude errors for different numbers of moving axles (linear shape function).

**Figure 16 sensors-23-05154-f016:**
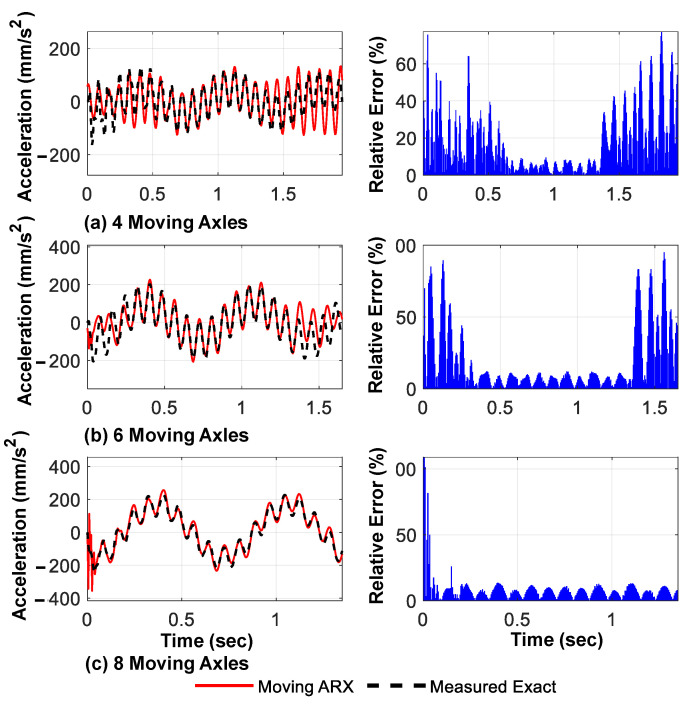
Predicted acceleration response of the mid-span using the proposed framework and their relative to amplitude errors for different numbers of moving axles (cubic spline shape function).

**Figure 17 sensors-23-05154-f017:**
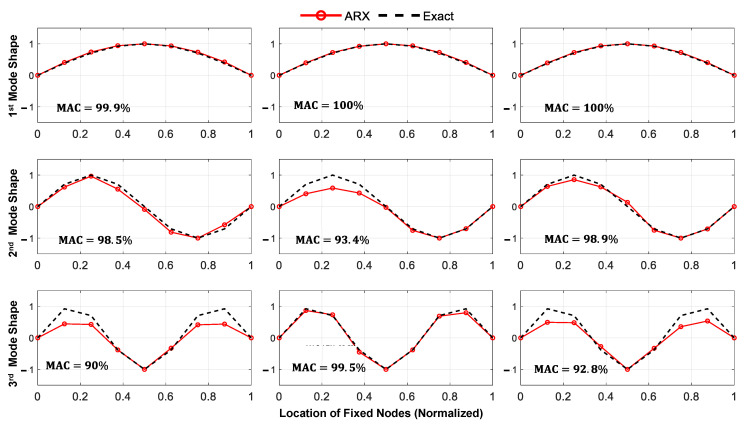
The first three identified mode shapes of the bridge under different number of moving axles (4, 6, and 8 moving axles) using the hybrid approach.

**Figure 18 sensors-23-05154-f018:**
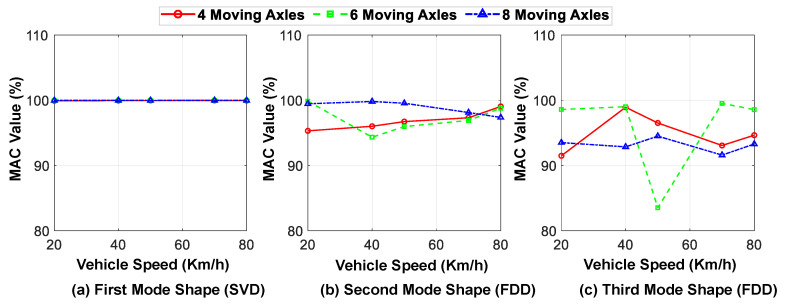
MAC values of the first three identified mode shapes at different speeds for different number of axles (in comparison with exact mode shapes).

**Figure 19 sensors-23-05154-f019:**
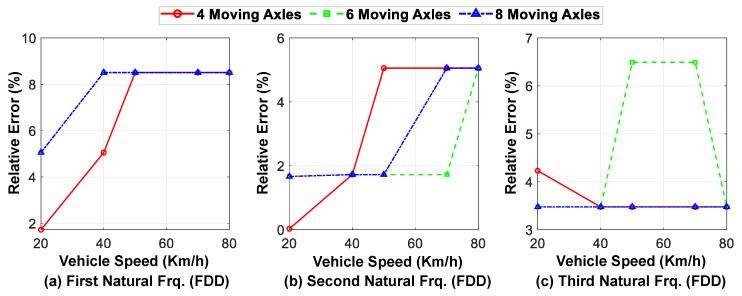
The relative error of the first three identified modal frequencies at different speeds for different numbers of axles (in comparison with exact natural frequencies).

**Figure 20 sensors-23-05154-f020:**
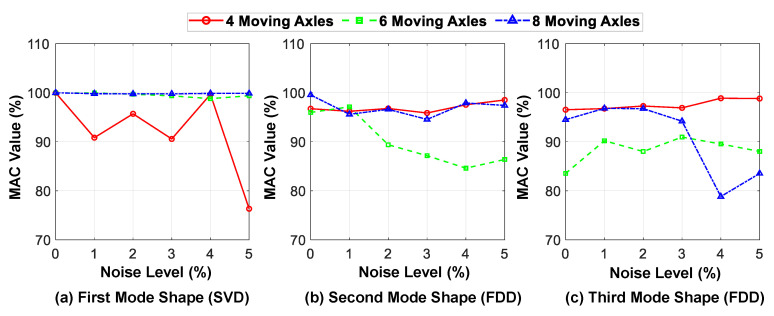
MAC values of the first three identified mode shapes at different noise levels for different numbers of axles (in comparison with exact mode shapes).

**Figure 21 sensors-23-05154-f021:**
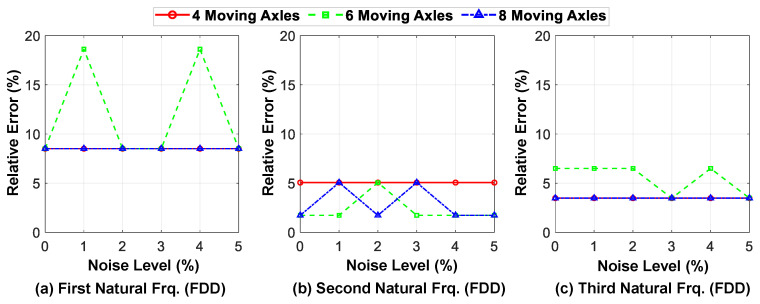
The relative errors of the first three identified modal frequencies at different noise levels for different numbers of axles (in comparison with exact values).

**Table 1 sensors-23-05154-t001:** List of symbols and notations used in this paper.

Symbol	Description
y(x,t)	Continuous function of bridge vertical displacement response
$y_{i}^{v}(t)$	Vertical displacement response of the *i*th axle of the vehicle
$D\left(t\right)$	Nodal vertical displacement vector of the bridge
Y(t)	Vector containing the displacement responses of the moving axles
N(x)	Interpolating shape function matrix for beam elements
$N_{v}(t)$	Interpolating shape function matrix for estimating the response of the moving axles (*m* × *n*)
$S_{j}$	*j*th virtual fixed node considered on the bridge
$N_{j}(x)$	Contribution of the *j*th node displacement of the displacement field
Q(t)	Vector containing the modal coordinate responses = [*q*_k_(*t*)]
$\Phi^{s}$	Matrix containing the amplitudes of all mode shapes at the fixed nodes
$\phi_{k}(s_{j})$	Amplitude of the *k*th mode shape at fixed node *S*_j_
*n*	Total number of virtual fixed nodes considered on the bridge (mesh nodes)
*m*	Total number of moving axles crossing the bridge
$s_{j}$	Location of the *j*th fixed node from the left support of the bridge
$x_{i}(t)$	Location of the *i*th moving axle from the left support of the bridge
Δs	Mesh size of the bridge element

**Table 2 sensors-23-05154-t002:** Identified natural frequencies and MAC values of the mode shapes.

Mode Number	Natural Frqs (Hz) (Exact)	Identified Natural Frqs (Hz) (FDD)	Error (%)	MAC Mode Shapes (%)
Mode 1	1.44	1.56	8.33	99.8
Mode 2	5.76	5.86	1.74	96.7
Mode 3	12.95	12.5	3.47	96.6

## Data Availability

Some or all data, models, or code that support the findings of this study are available from the corresponding author upon reasonable request.

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
