# Peer review of "A Mobile Sensing Framework for Bridge Modal Identification through an Inverse Problem Solution Procedure and Moving-Window Time Series Models"

_sensors, 2023, doi:10.3390/s23115154_

Round 1

Reviewer 1 Report

Some figures need to be prepared more carefully. For example, the labels of the Y-axes of the left-hand figures of Fig. 11(a) and (c) (Displacement (mm)), hide the values of the figure. Figure 10 does not explain what the red and green lines mean.

The section numbers used in the last paragraph of Introduction are not correct (not Section 3 but Section 2, for example). There is no Section 0 mentioned in the first sentence of 3.2.

It is necessary to explain more about the numerical simulations using ABAQUS in this study. More specific numerical model, governing equations, boundary conditions, material properties, have to be presented clearly so that the readers can reproduce and/or evaluate the results.

Author Response

Dear Editor and Reviewers,

Thank you for your valuable feedback on our manuscript titled "A Mobile Sensing Framework for Bridge Modal Identification through an Inverse Problem Solution Procedure and Moving-Window Time Series Models." We have carefully addressed all your comments and suggestions in the revised manuscript. We have provided a detailed response to each comment in a separate document, marked in red for clarity. Additionally, we have incorporated the suggested changes, including the addition of relevant references from the special issue. To facilitate the evaluation of our revisions, we are submitting the revised manuscript with the 'track changes' feature enabled. We believe that the revisions have significantly improved the quality and clarity of our paper, aligning it with the high standards of your journal. Thank you once again for your guidance throughout the review process. We appreciate the opportunity to contribute to your esteemed publication.

Reviewer 2 Report

Referee report for the manuscript sensors-2383323 entitled “A Hybrid Mobile Sensing Framework for Bridge Modal Identification through Inverse Problem Solving and Moving-Window Time Series Models”.

Dear Authors,

I read your paper and my comments, questions and suggestions are given below.

This paper presents a two-stage approach for determining the response signals of the virtual fixed nodes on the bridge and identifying the mode shapes and natural frequencies of the bridge. The method is based on the vibration response of vehicles passing over the bridge and a moving time series model approach. The proposed method was validated through numerical experiments which are based on models of a simple span bridge under the effect of moving mass. The effects of different levels of ambient noise, the number of axles of the passing vehicle, and the effect of vehicle speed on the accuracy of the method were also investigated. The results showed that the proposed method can identify the characteristics of the three main modes of the bridge with high accuracy. The paper fits to the scopes of the journal Sensors.

1. The authors name the developed method as a hybrid approach because it is based on Singular Value Decomposition (SVD) and Frequency Domain Decomposition (FDD). In my opinion, it is not reasonable enough. Could you explain more detailed the reasons behind the choice of this “hybrid” concept?

2. The abstract is fine. "Inverse problem" and "Vibration-based monitoring" should be added to Keywords.

3. The authors should check the last paragraph of Introduction. The order of sections mentioned here is not correct. The literature review should be improved. Try to avoid using references in the ways like that: [1–8], [9–16], [17–23], [21–26]. At least, some key ideas of previous studies should be given.

4. The proposed method is based on an inverse problem solution and the analysis of vibration response of vehicles passing over the bridge. These techniques are well known in the literature. Hence, in the abstract, sections of introduction and conclusions, the authors should clearly present the main contributions (or novelty) of your work, i.e. what’s the research gap that your study is trying to close. It is not well described in the current form.

5. Are all equations/formulas presented in the paper (Section 2) formed or proposed by the authors? If not then I suggest to provide proper references for them (where applicable).

6. Concerning the results shown in Figures 7 and 8, the authors implemented the iterative moving ARX framework to complete the response signal of fixed nodes. How many iterations were required to give the results in Figure 8(c)? I guess that you need only three iterations, but maybe I am wrong. The authors should mention this issue in the paper.

7. Section 2 in the current form is too long and difficult to follow. It consists of seven sub-sections. In my opinion, it should be re-written and re-organized. Some fundamental sub-sections can be grouped to form Section 2 (Background). The rest can be combined to from Section 3 which focuses only on presenting the proposed method.

8. Concerning the results in Figure 10, the linear shape function provides more precise displacement response estimates compared to the spline shape function. Could the authors give some more detailed explanations on this issue?

9. In Figures 18 and 20, can we conclude that the higher the order of the mode shape is, the better results we can obtain?

10. The section of conclusions should clearly highlight the major contributions of the study. Limitations of the proposed method should be also discussed in details here. More further works should be given. This part in the present form is likely a summary of the study.

11. The authors should check and correct the following minor mistakes: “Error! Reference source not found.” (line 275); “As explained in Section 0” (line 297); As it mentioned in section 0 (line 446).

The quality of English language is acceptable and understandable.

Author Response

(The authors gave the same response as above.)

Round 2

Reviewer 2 Report

The revised version of this paper has been significantly improved from the original submission; all of my comments/questions were adequately clarified and addressed. I do not have further comments. For me, the paper can be accepted.

The quality of English language is acceptable and understandable.